# The population affected by dust in China in the springtime

**Weijie Wang[1], Junjie Zhang[1,2]***

1 Environmental Research Center, Duke Kunshan University, Kunshan, Jiangsu, China, 2 Nicholas School of the Environment, Duke University, Durham, NC, United States of America

* junjie.zhang@duke.edu

**Data Availability Statement:** Data we use in this paper are available online and can be accessed using the following links. 10.6084/m9.figshare.21749567 10.6084/m9.figshare.21749573.

**Funding:** This study was supported by the National Natural Science Foundation of China (#71773043).

## Abstract

Dust events in northern China, particularly in the springtime, affect millions of people in the source and downwind regions. We investigate the population affected by various dust levels in China in the springtime from 2003 to 2020 using satellite retrievals of dust optical depth (DOD). We select three DOD thresholds, namely DOD > 0.2, DOD > 0.3, and DOD > 0.4, to estimate the population affected and find that each year the population affected can differ by one order of magnitude. The population exposed to DOD > 0.2 ranged from 16 million (2019) to over 200 million (2006). The population exposed to DOD > 0.3 ranged from 10 million (2015) to 70 million (2006). The population exposed to DOD > 0.4 ranged from 4 million (2017) to 36 million (2006). In years when dust events are frequent, people in the source and downwind regions are both affected, whereas, in years when dust events are less frequent, people affected are mainly in the source regions. Furthermore, we use the relative index of inequality to assess whether dust hazards impose unequal pollution burdens on different socioeconomic groups. We find that low-income communities have been more likely affected by dust pollution since 2013.

## 1. Introduction

Dust storms occur each spring in East Asia [1–4]. When meteorological conditions favor the mobilization and transport of dust, dust from source regions in northern China can travel thousands of kilometers downwind, causing severe air pollution [5,6] and affecting millions of people both in the source and downwind regions [7,8]. There are several major dust source regions in northern China, namely, the Taklamakan Desert in south Xinjiang and the Gobi Desert in Inner Mongolia [9–14]; together, they account for over 90% of the annual dust emissions in China [15]. Although dust storms are observed all year round in China, the Annual Book of Meteorological Disasters in China (2019) [16] reported that about two-thirds of these events occur in spring, particularly in March and April, when the frontal systems and the Mongolian cyclonic depression are active [9], favoring the emission and transport of dust in East Asia.

There exist numerous works focusing on dust storms in East Asia. Most works attempt to understand the physical aspects of dust storms, such as identifying dust source regions [11,17]

The funders had no role in study design, data collection and analysis, decision to publish, or preparation of the manuscript.

**Competing interests:** The authors have declared that no competing interests exist.

and land-use change in the source regions [18], illustrating the mechanisms that drive the formation of dust storms [19–21], and explaining the interannual variability of dust storm frequency [21–26]. In comparison, less work has been done to address the socioeconomic impacts of dust storms in China [27]. In addition, most studies on dust storms in China use ground-based observations from meteorological stations [9,10,24,25,28] scattered in less populated regions.

Dust is often mixed with local emissions, and both contribute to worsening air quality. A previous study found that 98.6% of the population in China was exposed to particulate matter at levels above World Health Organization's guideline [29]. After the outbreak of coronavirus disease, studies found that air quality improved in major cities in China in 2020 due to lockdown policies, and the reduction of particulate matter was significant [30–32]. However, most research on air pollution focuses on densely populated cities and provinces in China [33–35], and less attention has been paid to northwestern China, where ground-based observations are scarce. Thus, it is unclear how many people are affected by air pollution, especially dust, in northwestern China.

This work uses satellite observations to construct a dust climatology in China and estimate the population affected by dust each year. We focus on March and April as dust storms are more frequent in these two months. First, we construct a dust climatology using dust optical depth (DOD) from the Moderate Resolution Imaging Spectroradiometer (MODIS) to examine the spatial variability of dust in spring. Next, we examine the total population affected by various DOD levels, namely DOD > 0.2, DOD > 0.3, and DOD > 0.4, from 2003 to 2020. Furthermore, we examine the population affected by dust in each province during this period. This work aims to provide a detailed evaluation of the population affected by dust at the provincial level in China in the twenty-first century and to serve as a reference for health systems, preventive actions, and air quality control, especially for those provinces severely impacted by springtime dust.

## 2. Data and methods

### 2.1 MODIS DOD

MODIS instruments are aboard the Terra and Aqua satellites and have provided daily aerosol observations since 2000. We used the Deep Blue and Dark Target combined aerosol optical depth from the Aqua platform's level 3 daily aerosol products [36] at 1° longitude by 1° latitude spatial resolution to study dust over China. To eliminate other types of aerosols to obtain DOD, we applied the following three criteria based on previous works. First, the Angstrom exponent is less than 0.6 [37]. Since dust is mostly coarse particles, this criterion excludes fine aerosols [38]. Second, the single scattering albedo (SSA) is less than 0.95. This criterion effectively excludes sea salt in coastal regions as sea salt has an SSA close to 1 [39]. Third, the SSA at 412 nm is less than that at 670 nm as dust absorption increases from red to blue [39]. We used daily DOD from MODIS in March and April from 2003 to 2020 to evaluate the population affected by various levels of DOD.

### 2.2 Population and income

The annual population data of China is obtained from WorldPop (https://www.worldpop.org/) at 1km-by-1km resolution. We use this product to estimate the population affected by various levels of DOD from 2003 to 2020. The socioeconomic data is obtained from the China County Statistical Yearbooks (2003-2020). We use county gross domestic product (GDP) per capita as a proxy for local income. The county is the lowest-level jurisdiction that reports the GDP data. There were 2,844 counties in China by the end of 2020.

## 2.3 Inequality of dust hazard

We analyze the unequal distribution of dust hazards across communities with different income levels visually and numerically. Let $p$ index the cumulative share of the population ranked by income levels. The concentration curve, $D(p)$, designates the cumulative share of dust hazard. We can measure the inequality of dust hazard by the concentration index (CI), which is twice the area between the concentration curve and the diagonal:

$$CI = 1 - 2 \int_0^1 D(p)dp. \tag{1}$$

If the dust hazard is perfectly equal, the concentration curve should coincide with the diagonal ($CI = 0$). If the curve lies above the diagonal ($CI<0$), low-income areas are more exposed to the dust hazard; otherwise, the opposite case would be true for $CI>0$.

Following Kakwani et al. [40], we use the relative index of inequality (RII) to estimate the unequal distribution of dust across communities with different levels of income. First of all, we rank the county by GDP per capita each year. For county $i$ ($i = 1, \ldots, I$), let $c_i$ designate its dust concentration and $p_i$ for its population share. The average dust hazard is then $c = \sum_{i=1}^I p_i c_i$. The relative rank of county $i$ in terms of GDP per capita is $r_i = \sum_{j=1}^{i-1} p_j + p_i/2$. Then we run the following regression:

$$\frac{c_i}{c} \sqrt{p_i} = \beta_1 \sqrt{p_i} + \beta_2 r_i \sqrt{p_i} + \varepsilon_i. \tag{2}$$

In this form, $\beta_1$ and $\beta_2$ are parameters to be estimated, and $\varepsilon_i$ is an unobservable error term. Please note that $\beta_2$ measures the relative index of inequality. The CI can be derived from the RII through $\hat{CI} = 2\hat{\beta}_2 \sum_{i=1}^I p_i(r_i - 1/2)$, where $\hat{CI}$ designates the estimated concentration index.

## 3. Results

### 3.1 Springtime dust climatology and time series

DOD exhibits significant spatial variability in China. Fig 1A shows March-April mean DOD climatology from 2003 to 2020 using MODIS. It indicates that the Taklamakan Desert in Xinjiang is a major dust source region, with DOD above 0.6 in March and April, and consequently, DOD in Xinjiang is the highest in the springtime, with March-April mean DOD of 0.36. March-April mean DOD is also high in provinces such as Qinghai (0.22), Ningxia (0.25), Gansu (0.20), and Inner Mongolia (0.12). Within the Gobi Desert, an area called the Hexi Corridor extends from 105 to 110˚E and 38 to 42˚N, where the topography allows the passage of dust from source regions to downwind regions [3]. Although DOD is the highest over the Taklamakan Desert, dust is mostly trapped in the Tarim Basin, while dust originating from the Gobi Desert is mainly transported to downwind regions. March-April mean DOD in central China provinces, such as Shanxi and Hebei, is around 0.06, while it is below 0.05 in coastal areas, such as Shandong and Jiangsu (Fig 1A). The springtime DOD clearly shows that DOD is the highest in the source region in Xinjiang, decreases in the downwind provinces in central China, and further decreases in eastern China, where the impact of dust usually is less felt (Fig 1A). Springtime dust is also observed in northeastern China, including Heilongjiang, Jilin, and Liaoning, as dust from source regions in eastern Inner Mongolia and Gobi Deserts is advected downwind towards these three provinces (Fig 1A). March-April mean DOD in these three provinces is between 0.06 and 0.09.

Dust events also exhibit interannual variability. Fig 1B shows March-April mean DOD over China using MODIS from 2003 to 2020 and the frequency of dust events recorded at

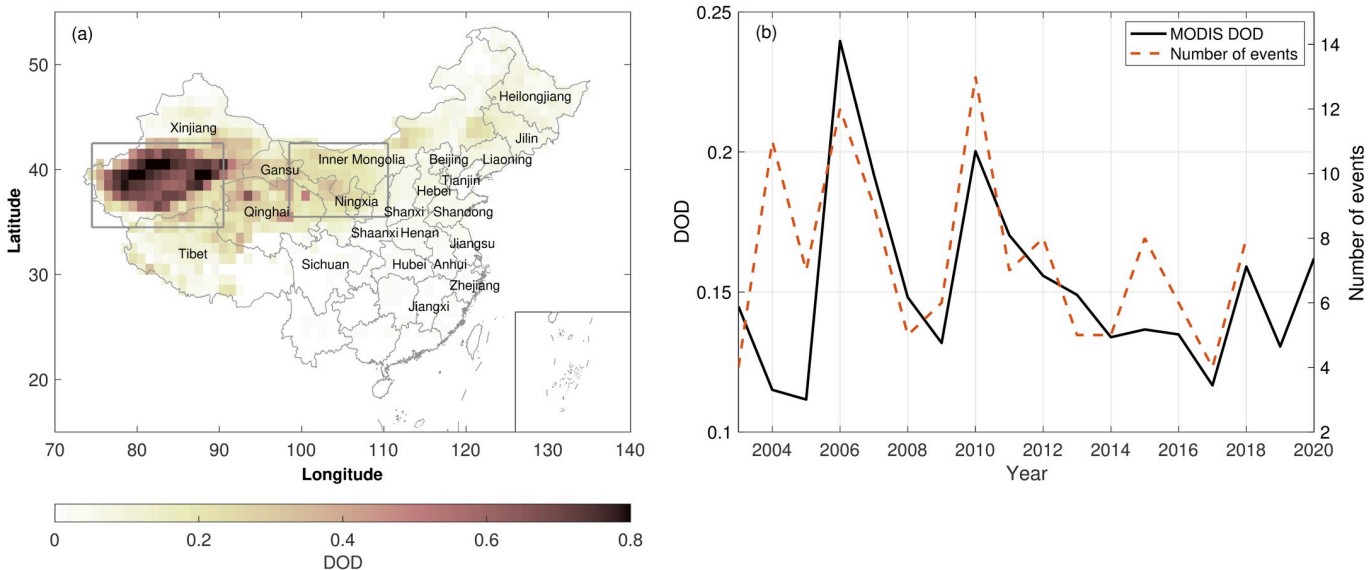

**Fig 1.** (a) March-April mean DOD at 1° by 1° over China from 2003 to 2020 using MODIS daily DOD. The gray boxes are the major dust source regions of the Taklamakan Desert (74.5 to 90.5°E, 34.5 to 42.5°N) and the Gobi Desert (98.5 to 110.5°E, 35.5 to 42.5°N). (b) March-April mean DOD over China from 2003 to 2020 using MODIS (solid line) and the total number of dust events in March and April in China (dashed line) reported by the Annual Book of Meteorological Disasters in China (2019).

meteorological stations from 2003 to 2018. In the past 18 years, dust events were most frequent in March and April in 2010, with a total number of thirteen events, and least frequent in 2003 and 2017, with a total number of four events in March and April, as reported by the Annual Book of Meteorological Disasters in China (2019). In most years, DOD from MODIS agrees with the frequency of dust events observed at meteorological stations (Fig 1B). For example, MODIS showed the highest mean DOD of 0.24 in 2006, and the frequency of dust events observed this year is the second-highest during the study period, while in a year when dust events are less frequent, such as 2017, the mean DOD is relatively low (Fig 1B). However, it should be noted that a higher frequency of dust events does not necessarily indicate a higher mean DOD, as dust events may vary in strength and area of impact, and vice versa. Take 2004 as an example. The number of dust events is the third highest while DOD in the second lowest over two decades (Fig 1B). In this work, we use satellite observations to estimate the population affected by dust as they provide better spatial coverage and thus would perform better than ground-based observations.

## 3.2 The total population affected by dust

Dust events can cause severe air pollution in the source and downwind regions. Since China's total population exceeded 1.4 billion in 2020, a dust event can cause severe public health consequences by affecting millions of people. To estimate the population affected by various levels of DOD each year, we select three DOD thresholds, namely, DOD > 0.2, DOD > 0.3, and DOD > 0.4, using MODIS. Fig 2A shows the population exposed to DOD above these three thresholds from 2003 to 2020. Overall, the population affected in a year with a higher mean DOD is likely to be larger (Fig 2B). For example, the highest March-April mean DOD over China is in 2006 (Fig 1B), and the population affected by dust is also most extensive in the same year (Fig 2A). The population exposed to DOD above 0.2 is estimated to be 208.4 million

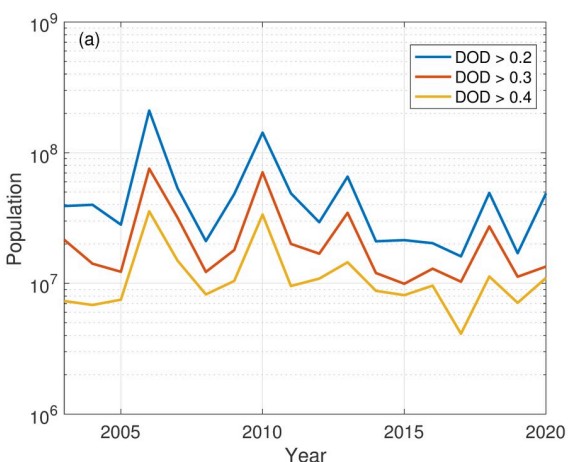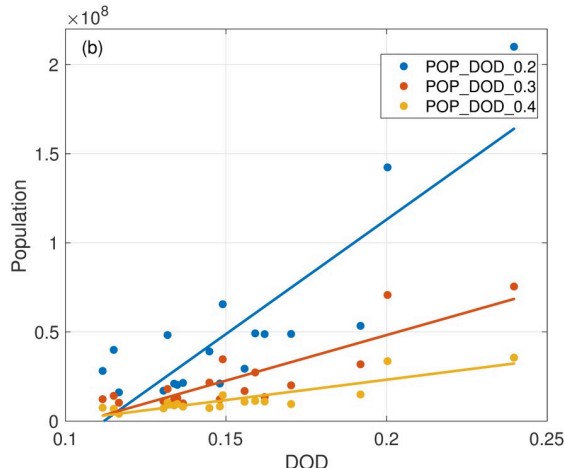

**Fig 2.** (a) The population affected by March-April mean DOD above 0.2 (blue), above 0.3 (red), and above 0.4 (yellow) from 2003 to 2020 estimated using MODIS. The y-axis is on a log scale. (b) The population affected by DOD above 0.2 (POP_DOD_0.2), the population affected by DOD above 0.3 (POP_DOD_0.3), and the population affected by DOD above 0.4 (POP_DOD_0.4) as a function of DOD. Straight lines are the linear regression fit. POP_DOD_0.2 = $1.28 \times 10^9$DOD - $1.44 \times 10^8$ ($r^2$ = 0.73), POP_DOD_0.3 = $5.11 \times 10^8$DOD - $5.39 \times 10^7$ ($r^2$ = 0.74), POP_DOD_0.4 = $2.29 \times 10^8$DOD - $2.25 \times 10^7$ ($r^2$ = 0.76).

(S1 Table), accounting for roughly 16% of the total population of China at that time. The population affected by DOD above 0.3 in the same year is estimated to be 75.2 million (Fig 2A, S2 Table), and the population affected by DOD above 0.4 is 35.5 million (Fig 2A, S3 Table). The year 2010 has the second-highest mean DOD (Fig 1B), and the population affected by dust is also the second-largest (Fig 2A). The population affected by DOD above 0.2, 0.3, and 0.4 in 2010 is estimated to be 142.2 million, 70.7 million, and 33.6 million, respectively (Fig 2A, S1–S3 Tables). In comparison, in a year with a lower mean DOD, such as 2017 and 2019, the population affected is about an order of magnitude smaller than in a year of a higher mean DOD, such as 2006 and 2010 (Figs 1B and 2A, S1–S3 Tables). The population affected by DOD above 0.2, 0.3, and 0.4 in 2017 are estimated to be 16.1 million, 10.3 million, and 4.0 million, respectively (Fig 2A, S1–S3 Tables). The population affected by the three thresholds of DOD from 2003 to 2020 is summarized in S1–S3 Tables.

### 3.3 Spatial heterogeneity of dust hazards

The population in China exhibits vast spatial variability, with 94% of the people living east of the Heihe-Tengchong line (Fig 1A), and is much smaller in northern and northwestern China, where major dust source regions are located (Figs 1A and 3). Thus, the population affected by dust varies significantly in each province. Fig 3B shows locations with March-April mean DOD above 0.2 averaged over 2003 and 2020. It can be seen that five provinces, Xinjiang, Gansu, Qinghai, Inner Mongolia, and Tibet, have locations with March-April mean DOD above 0.2. Those five provinces also have locations with March-April mean DOD above 0.3 (Fig 3C). Locations with March-April mean DOD above 0.4 are only found in Xinjiang and Qinghai (Fig 3D).

To examine the spatial variability of the population exposed to various levels of DOD, we compare two years with relatively high and low mean DOD, namely 2006 and 2017. Fig 4 shows the March-April mean DOD and the population affected by dust in each province in 2006. When dust events are frequent, locations with March-April mean DOD above 0.2 are found in 17 provinces, including all provinces in northwestern China (Xinjiang, Qinghai, Gansu, Ningxia, and Shaanxi), northern China (Inner Mongolia, Hebei, and Shanxi), and

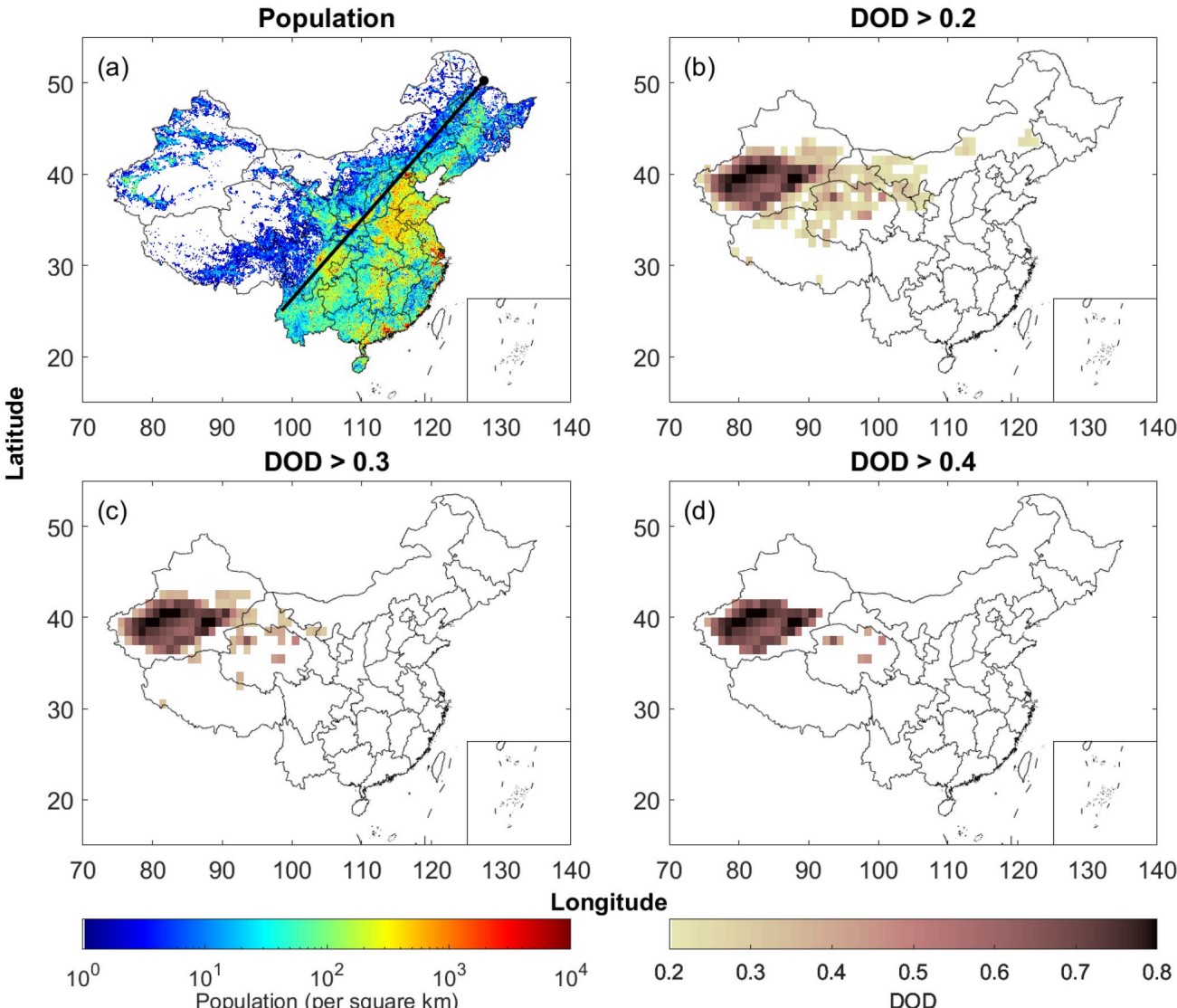

**Fig 3.** (a) Population per square kilometer in China in 2020 from Worldpop. The black line is the Heihe-Tengchong line. Locations with March-April mean DOD above (b) 0.2, (c) 0.3, and (d) 0.4 over China from 2003 to 2020 using MODIS daily DOD.

northeastern China (Heilongjiang, Jilin, and Liaoning) (Fig 4A). The capital city Beijing and the Municipality of Tianjin are also affected (Fig 4A). Shandong is the most affected province, followed by Hebei (Fig 4B). The population affected by DOD above 0.2 in 2006 in Shandong and Hebei is estimated to be 37.8 million and 34.9 million, respectively (S1 Table). Although both provinces are located in the downwind regions, the population affected by dust is larger compared with the source regions as the population density is larger in these two provinces (Fig 3). In addition, there are six provinces and one municipality where the population exposed to DOD above 0.2 is over 10 million, namely, Inner Mongolia (15.8 million), Xinjiang (10.5 million), Gansu (21.8 million), Heilongjiang (12.3 million), Liaoning (26.0 million), Jilin (17.0 million), and Tianjin (10.6 million) (S1 Table). The three provinces in northeastern China can also be affected by dust from the nearby Songnen Plain and dust from Inner Mongolia [41]. Locations with March-April mean DOD above 0.3 are found in 12 provinces in

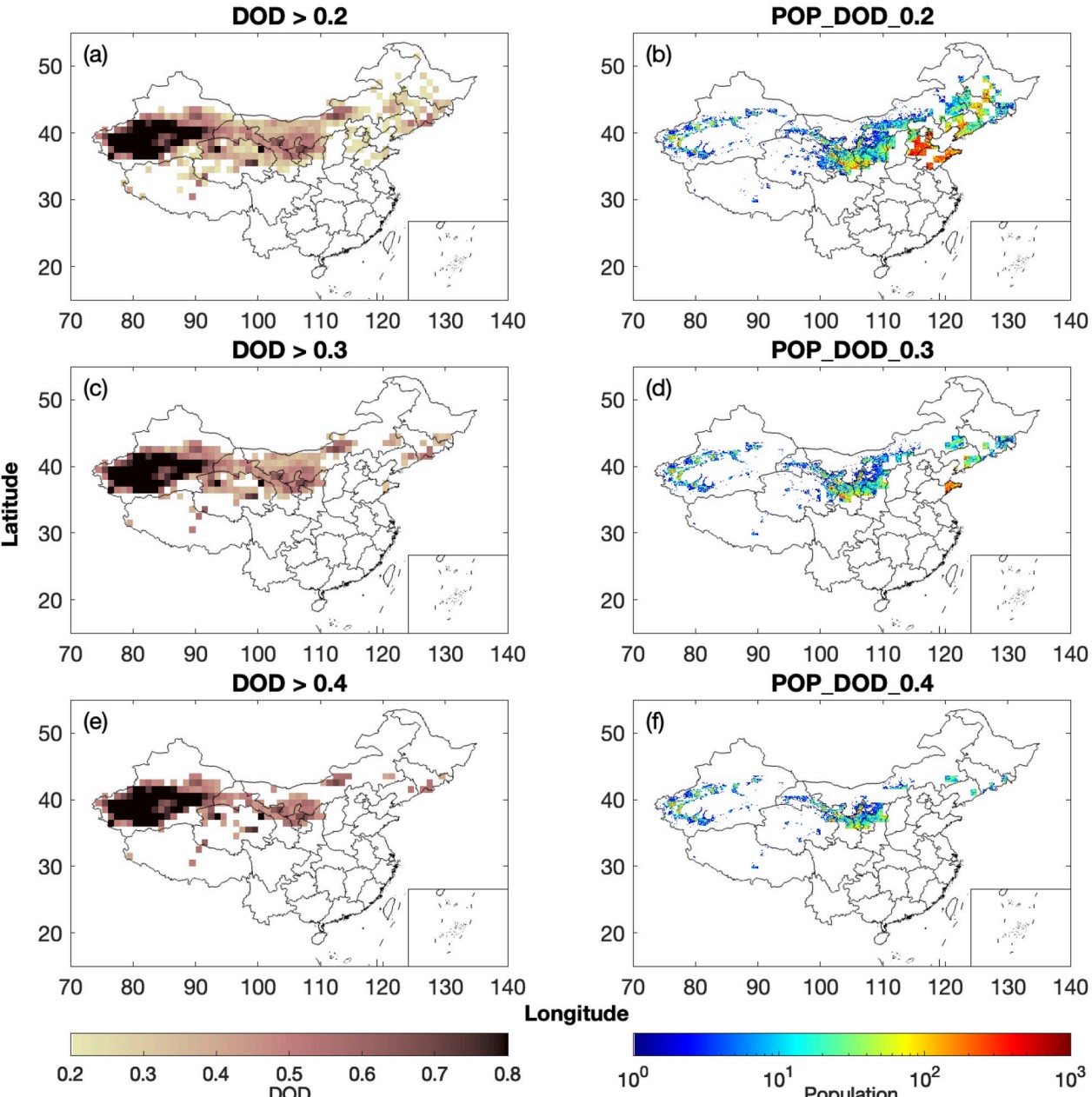

**Fig 4.** Locations with March-April mean DOD above (a) 0.2, (c) 0.3, and (e) 0.4 in 2006 using MODIS and the spatial distribution of the population affected by DOD above (b) 0.2, (d) 0.3, and (f) 0.4 in 2006.

2006, including all provinces in northwestern, northern, and northeastern China (Fig 4C). The population affected by DOD above 0.3 is the largest in Gansu (Fig 4D) and is estimated to be 15.9 million (S2 Table). The population affected by DOD above 0.3 is above 10 million in Shandong (14.1 million) and Xinjiang (10.4 million) and below 10 million in other provinces in 2006 (S2 Table). Locations with March-April mean DOD above 0.4 are found in 11 provinces in 2006 (Fig 4E). The population affected by DOD above 0.4 is the largest in Xinjiang (Fig 4F) and is estimated to be 10.1 million, followed by Gansu, where 8.4 million people are affected by mean DOD above 0.4 (S3 Table).

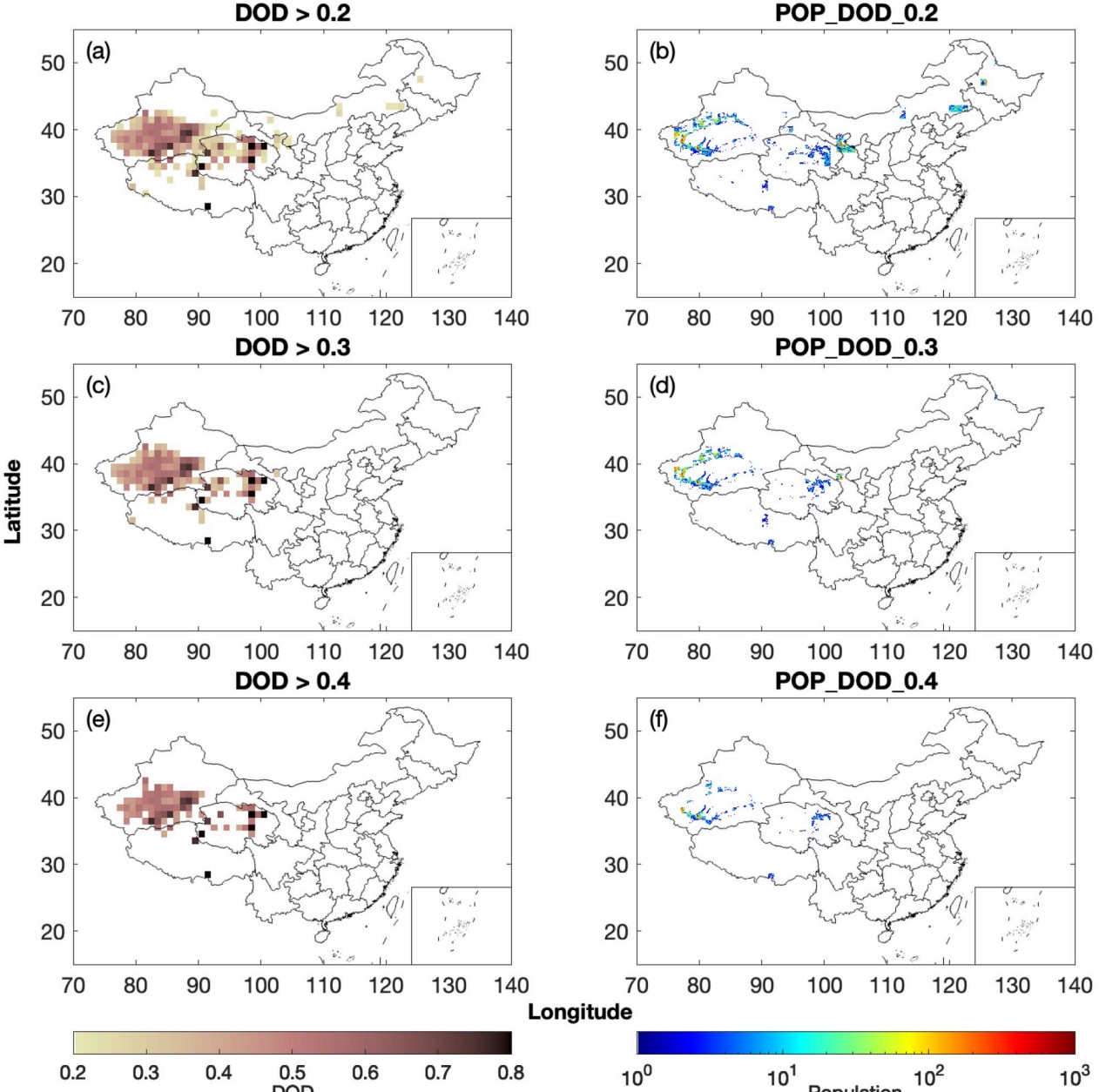

**Fig 5.** Locations with March-April mean DOD above (a) 0.2, (c) 0.3, and (e) 0.4 in 2017 using MODIS and the spatial distribution of the population affected by DOD above (b) 0.2, (d) 0.3, and (f) 0.4 in 2017.

In comparison, fewer people are exposed in each province when March-April mean DOD is relatively low. Fig 5 shows the March-April mean DOD and the population affected in each province in 2017. Locations with March-April mean DOD above 0.2 are found in six provinces, including four provinces with dust sources, namely Xinjiang, Qinghai, Gansu, and Inner Mongolia (Fig 5A). Locations with March-April mean DOD above 0.3 are found in four provinces, among which three are in northwestern China with dust sources (Fig 5C). Locations with March-April mean DOD above 0.4 are found only in two provinces, Xinjiang and Qinghai (Fig 5E). The population affected by dust in 2017 is mainly in Xinjiang (Fig 5B, 5D and

5F), where the population affected by DOD above 0.2, 0.3, and 0.4 are 9.1 million, 8.3 million, and 3.8 million, respectively (S1–S3 Tables). The population affected by dust in Gansu is the second largest, with the population affected by DOD above 0.2 and 0.3 being 3.1 million and 1.4 million, respectively (S1 and S2 Tables). The population affected by various levels of DOD in each province each year is summarized in S1–S3 Tables. The mean DOD and population affected by dust levels for years other than 2006 and 2017 are given in S1–S16 Figs.

## 3.4 Income disparities in dust hazards

The heterogeneous distribution of dust events and dust concentration likely imposes unequal pollution burdens on different socioeconomic groups. We are concerned about whether low-income communities are more susceptible to dust pollution. To test this hypothesis, we use the concentration curve to illustrate the inequality of dust hazards. First, we rank the counties by GDP per capita each year, from the lowest-income county to the highest-income county. Then we plot the cumulative share of dust exposure against the cumulative share of the population ranked by income, one curve for each year. Fig 6(A) shows that the concentration curve fluctuates around the diagonal, suggesting there is no clear relationship between dust pollution and income.

We also use the relative index of inequality (RII) to estimate the unequal distribution of dust across the communities with different income levels, following Kakwani et al. [40]. The RII is the slope of the regression of a county's relative dust hazard on its relative income rank, weighted by the population share (see data and methods). We estimate the RII for each year. The point estimates and confidence intervals are summarized in Fig 6B). It shows that the inequality of dust hazards exhibits strong temporal heterogeneity. There are mainly two periods with different effects. During 2004-2012, dust hazards have an ambiguous inequality

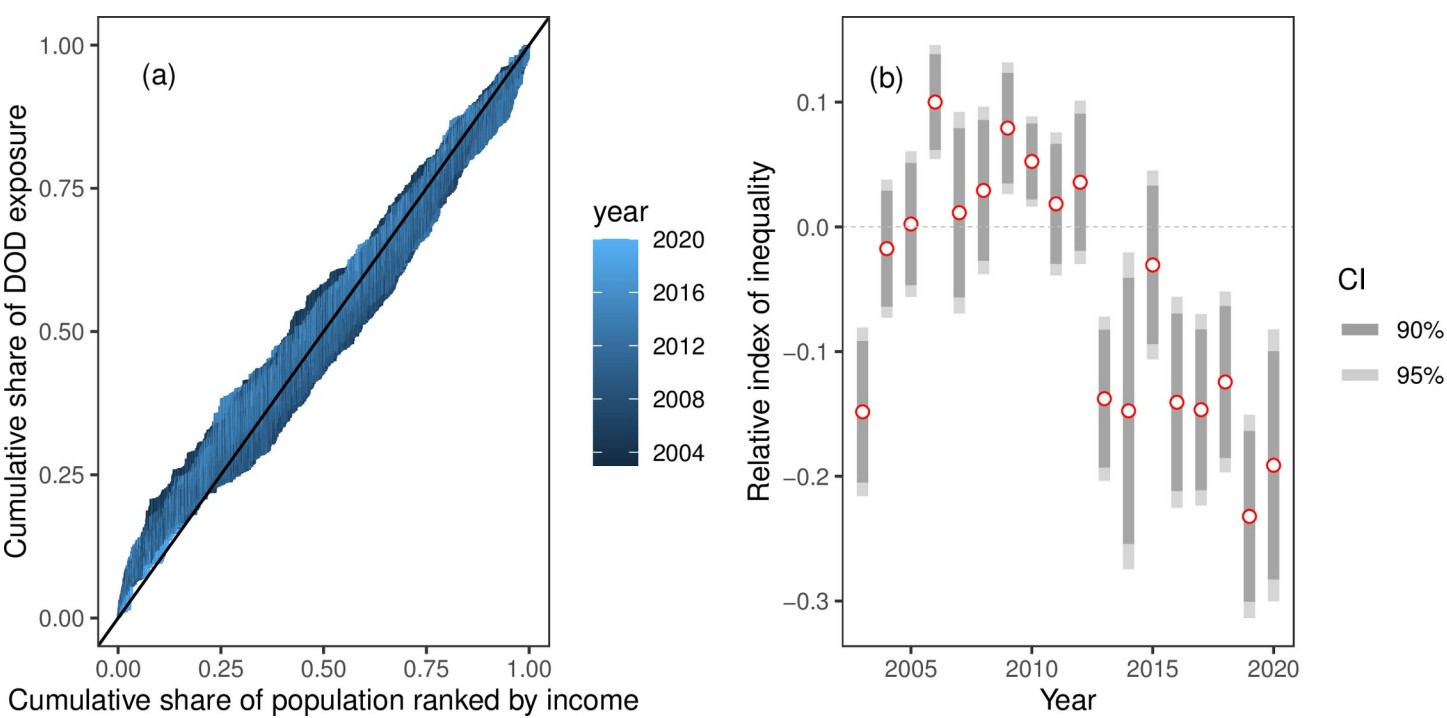

**Fig 6.** (a) Concentration curve depicting the cumulative share of dust exposure against the cumulative share of pollution. One curve for each year. (b) The estimated relative index of inequality for each year. The shades represent the confidence intervals with 90% or 95% of confidence levels.

implication. Most estimated RIIs in this period are statistically insignificant except for 2006 and 2009, suggesting that high-income regions were affected by dust. After 2013, low-income regions were affected by dust more frequently; most estimated RIIs are statistically significant except for 2015. Such analysis shows that the impact of dust is not related to income levels.

## 4. Discussion

### 4.1 Satellite and meteorological observations

Previous studies often used dust events frequency and dust concentration observed at meteorological stations to study the intensity of dust events. Dust events are classified into four categories based on visibility, dust in suspension, blowing dust, dust storm, and severe dust storm. Dust event intensity is defined as a function of the dust concentration observed during these four events [23,25]. However, there are two problems with using observations from meteorological stations to estimate the impact of dust events. First, the number of stations is limited in less populous regions, and there are barely any stations at the center of the Taklamakan Desert, where DOD is highest (Fig 1A). Thus, using observations from the limited number of stations in the surrounding regions of the Taklamakan Desert might underestimate the intensity of dust events in the source regions. Second, the spatial coverage of observations made at meteorological stations is also limited. Although geospatial interpolation of ground-based data is used to study the spatial distribution of dust [26], such distribution of dust may not represent the real-world situation as meteorological conditions, such as wind speed and direction, are not taken into consideration. In comparison, satellite observations can better represent the intensity of dust events and the spatial distribution of dust. Our analysis indicates that a higher number of dust events does not necessarily correspond to a higher mean DOD. For example, a total number of 11 dust events occurred in 2004, but the mean DOD is 0.12, lower than in 2003, when only four dust events occurred (Fig 1B). However, satellite observations also have caveats. For example, there might be missing data due to the existence of cloud cover. In addition, DOD derived from satellite represents the dust in the total column, while humans are only exposed to dust near the surface, and the relationship between DOD and surface dust concentration has some uncertainty. Nevertheless, satellite observations can complement ground-based observations in estimating the impact of dust on a large scale.

### 4.2 The most affected provinces

The spatial distribution of dust from satellite observations makes it possible to estimate the population exposed to various dust levels. On average, Xinjiang is the province with the most people affected by dust over the past 18 years (S1–S3 Tables). The population affected by March-April mean DOD above 0.2, 0.3, and 0.4 in Xinjiang averaged over 2003 - 2020 is 11.4 million, 9.3 million, and 7.9 million, respectively. Gansu is the province with the second-most people affected by dust (S1–S3 Tables). The population affected by March-April mean DOD above 0.2, 0.3, and 0.4 in Gansu averaged over 2003 and 2020 is 9.8 million, 5.5 million, and 1.7 million, respectively. Inner Mongolia is the province with the third-most people affected (S1–S3 Tables). The population affected by March-April mean DOD above 0.2, 0.3, and 0.4 in Inner Mongolia averaged over 2003, and 2020 is 4.7 million, 1.5 million, and 0.4 million, respectively. Since the two major dust source regions are Xinjiang and Inner Mongolia, and Gansu is adjacent to Xinjiang and Inner Mongolia, it can be expected that, on average, the population affected by dust is the largest in these three provinces. In addition to the three provinces discussed above, there are nine other provinces where the population affected by March-April mean DOD above 0.2 exceeds one million, and most of these provinces are in northwestern, northern, and northeastern China (S1 Table). Qinghai and Ningxia are the two other

provinces where the population affected by March-April mean DOD above 0.3 exceeds one million, in addition to Xinjiang, Gansu, and Inner Mongolia (S2 Table).

The population affected by dust is mostly in northwestern, northern, and northeastern China. In comparison, the pollution affected by other types of pollutants, such as $SO_2$, $NO_2$, and $O_3$, is mostly in megacities and surrounding areas, such as Beijing-Tianjin-Hebei and the Yangtze River Delta [34]. The population affected by air pollution in China was estimated to be 975 million in 2010 [42], whereas the population affected by DOD > 0.2 was 142 million. It is worth noting that the criterion used by Liu et al. (2007) is different from that used in this work. Nevertheless, dust contributes significantly to particulate matter pollution.

## 4.3 Health impact of dust

These provinces in northwestern and northern China, affected by dust most severely, are less economically developed than coastal provinces. The per capita GDP in Gansu was $5,223 in 2020, about a quarter of that in coastal provinces such as Jiangsu ($19,230). The per capita GDP in Xinjiang ($8,250) and Inner Mongolia ($10,320) were roughly half that in Jiangsu in 2020. Thus, people living in dust-affected regions may be less willing to pay for preventive expenditures related to air pollution due to economic factors [43]. In addition, healthcare facilities are less accessible due to the lack of facilities in rural regions [44]. After the health care reform in 2009, studies have found that although health resources have increased after the reform, inequalities in health resources between the wealthiest and the poorest counties in rural China have also increased, with wealthier counties getting more health resources [45]. Healthcare expenditures are lower in rural areas due to low income [44,46]. Consequently, people in northwestern and northern China, especially those in low-income counties, may have a higher risk of respiratory diseases associated with air pollution.

Studies have shown that emergency room visits for respiratory diseases increased on dust days in the source region [47,48] and downwind regions [49,50]. Meanwhile, respiratory mortality increases in days after a dust event [51]. It has been suggested that a dust early warning system can help improve public health [52]. Since dust emission is associated with high wind speed and dust transport from source to downwind regions usually takes several days, it is possible to forecast a dust event. For example, the Coupled Ocean Atmosphere Mesoscale Prediction System produces three-day forecasts of dust for Southwest Asia and has successfully predicted more than 85% of the observed dust events in Iraq during the study period [53]. If an early warning is issued, those in dust-affected regions can choose to wear masks and stay indoors. Such precautions may reduce health risks, especially for the elderly and children.

## 4.4 Future trends of dust

Dust events frequency in China has exhibited a downward trend over the past half-century mainly due to the reduction of winds [21,25], and it has been predicted that such a trend would continue in the twenty-first century [54]. Over the past two decades, March-April mean DOD exhibits a downward trend (Fig 1B), and the population affected by dust also decreases despite the natural increase of the population over this period (Fig 2A). Given the fact that dust emissions would fall and the population in China would likely decrease beyond 2020, it is possible that the population affected by dust would drop in the future. However, it is still necessary to understand the frequency and intensity of dust events in China in the future. Although the population affected may decrease, it is on the order of millions. Therefore, it is helpful to understand the future trend of dust activities so that preventive actions can be taken to reduce losses related to extreme dust events and health risks associated with air pollution.

## 5. Conclusions

To summarize, the population affected by dust in the springtime varies each year in China, and people in the source regions and downwind regions are both affected. In general, mean DOD is positively correlated with the population affected by dust. Over the past 18 years, March-April mean DOD over China was highest in 2006, and the population affected by DOD above 0.2 is estimated to be 208.4 million. The population affected by DOD above 0.2 in 2006 is more extensive in downwind regions due to larger population density, while the population affected by DOD above 0.3 and 0.4 are larger in source regions as higher DOD is mainly found there. Overall, people in eastern China are less likely to be affected by dust, but in extreme cases, they can also be exposed to a mean DOD above 0.2. In comparison, the population affected by dust is an order of magnitude smaller in a year with low March-April mean DOD, and the population exposed to various dust levels is larger in the source regions, while the population affected in downwind regions is small. This study offers details about the population affected by dust at a provincial level. It may have implications for health risk assessment, especially for those provinces with a higher concentration of dust in the springtime.

## Supporting information

**S1 Fig.** Locations with March-April mean DOD above (a) 0.2, (c) 0.3, and (e) 0.4 in 2003 using MODIS and the spatial distribution of the population affected by DOD above (b) 0.2, (d) 0.3, and (f) 0.4 in 2003.
(TIF)

**S2 Fig.** Locations with March-April mean DOD above (a) 0.2, (c) 0.3, and (e) 0.4 in 2004 using MODIS and the spatial distribution of the population affected by DOD above (b) 0.2, (d) 0.3, and (f) 0.4 in 2004.
(TIF)

**S3 Fig.** Locations with March-April mean DOD above (a) 0.2, (c) 0.3, and (e) 0.4 in 2005 using MODIS and the spatial distribution of the population affected by DOD above (b) 0.2, (d) 0.3, and (f) 0.4 in 2005.
(TIF)

**S4 Fig.** Locations with March-April mean DOD above (a) 0.2, (c) 0.3, and (e) 0.4 in 2007 using MODIS and the spatial distribution of the population affected by DOD above (b) 0.2, (d) 0.3, and (f) 0.4 in 2007.
(TIF)

**S5 Fig.** Locations with March-April mean DOD above (a) 0.2, (c) 0.3, and (e) 0.4 in 2008 using MODIS and the spatial distribution of the population affected by DOD above (b) 0.2, (d) 0.3, and (f) 0.4 in 2008.
(TIF)

**S6 Fig.** Locations with March-April mean DOD above (a) 0.2, (c) 0.3, and (e) 0.4 in 2009 using MODIS and the spatial distribution of the population affected by DOD above (b) 0.2, (d) 0.3, and (f) 0.4 in 2009.
(TIF)

**S7 Fig.** Locations with March-April mean DOD above (a) 0.2, (c) 0.3, and (e) 0.4 in 2010 using MODIS and the spatial distribution of the population affected by DOD above (b) 0.2, (d) 0.3, and (f) 0.4 in 2010.
(TIF)

**S8 Fig.** Locations with March-April mean DOD above (a) 0.2, (c) 0.3, and (e) 0.4 in 2011 using MODIS and the spatial distribution of the population affected by DOD above (b) 0.2, (d) 0.3, and (f) 0.4 in 2011.
(TIF)

**S9 Fig.** Locations with March-April mean DOD above (a) 0.2, (c) 0.3, and (e) 0.4 in 2012 using MODIS and the spatial distribution of the population affected by DOD above (b) 0.2, (d) 0.3, and (f) 0.4 in 2012.
(TIF)

**S10 Fig.** Locations with March-April mean DOD above (a) 0.2, (c) 0.3, and (e) 0.4 in 2013 using MODIS and the spatial distribution of the population affected by DOD above (b) 0.2, (d) 0.3, and (f) 0.4 in 2013.
(TIF)

**S11 Fig.** Locations with March-April mean DOD above (a) 0.2, (c) 0.3, and (e) 0.4 in 2014 using MODIS and the spatial distribution of the population affected by DOD above (b) 0.2, (d) 0.3, and (f) 0.4 in 2014.
(TIF)

**S12 Fig.** Locations with March-April mean DOD above (a) 0.2, (c) 0.3, and (e) 0.4 in 2015 using MODIS and the spatial distribution of the population affected by DOD above (b) 0.2, (d) 0.3, and (f) 0.4 in 2015.
(TIF)

**S13 Fig.** Locations with March-April mean DOD above (a) 0.2, (c) 0.3, and (e) 0.4 in 2016 using MODIS and the spatial distribution of the population affected by DOD above (b) 0.2, (d) 0.3, and (f) 0.4 in 2016.
(TIF)

**S14 Fig.** Locations with March-April mean DOD above (a) 0.2, (c) 0.3, and (e) 0.4 in 2018 using MODIS and the spatial distribution of the population affected by DOD above (b) 0.2, (d) 0.3, and (f) 0.4 in 2018.
(TIF)

**S15 Fig.** Locations with March-April mean DOD above (a) 0.2, (c) 0.3, and (e) 0.4 in 2019 using MODIS and the spatial distribution of the population affected by DOD above (b) 0.2, (d) 0.3, and (f) 0.4 in 2019.
(TIF)

**S16 Fig.** Locations with March-April mean DOD above (a) 0.2, (c) 0.3, and (e) 0.4 in 2020 using MODIS and the spatial distribution of the population affected by DOD above (b) 0.2, (d) 0.3, and (f) 0.4 in 2020.
(TIF)

**S17 Fig. Concentration curve for the dust exposure.**
(TIF)

**S1 Table. The population (in a million) affected by March-April mean DOD $> 0.2$ in each province from 2003 to 2020.**
(DOCX)

**S2 Table. The population (in a million) affected by March-April mean DOD $> 0.3$ in each province from 2003 to 2020.**
(DOCX)

**S3 Table. The population (in a million) affected by March-April mean DOD > 0.3 in each province from 2003 to 2020.**
(DOCX)

## Author Contributions

**Conceptualization:** Weijie Wang, Junjie Zhang.

**Data curation:** Weijie Wang.

**Formal analysis:** Weijie Wang, Junjie Zhang.

**Funding acquisition:** Junjie Zhang.

**Investigation:** Weijie Wang, Junjie Zhang.

**Methodology:** Weijie Wang, Junjie Zhang.

**Project administration:** Junjie Zhang.

**Resources:** Weijie Wang, Junjie Zhang.

**Software:** Weijie Wang, Junjie Zhang.

**Supervision:** Junjie Zhang.

**Validation:** Weijie Wang, Junjie Zhang.

**Visualization:** Weijie Wang, Junjie Zhang.

**Writing – original draft:** Weijie Wang, Junjie Zhang.

**Writing – review & editing:** Weijie Wang, Junjie Zhang.

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
