## [Decision Letter · Decision Letter 0]

12 Dec 2022

PONE-D-22-31637The Population Affected by Dust in China in the SpringtimePLOS ONE

Dear Dr. Zhang,

Thank you for submitting your manuscript to PLOS ONE. After careful consideration, we feel that it has merit but does not fully meet PLOS ONE’s publication criteria as it currently stands. Therefore, we invite you to submit a revised version of the manuscript that addresses the points raised during the review process.

We look forward to receiving your revised manuscript.

Kind regards,

Uzair Aslam Bhatti

Academic Editor

PLOS ONE

 “This study was supported by the National Natural Science Foundation of China (#71773043).” 

“This study was supported by the National Natural Science Foundation of China (#71773043).”

“This study was supported by the National Natural Science Foundation of China (#71773043).” 

5. We note that Figures 1,3,4,5,s4,s5,s6,s7,s8,s9,s10,s11,s12,s13,s14,s15,s16,s17,s18 and s19 in your submission contain [map/satellite] images which may be copyrighted. All PLOS content is published under the Creative Commons Attribution License (CC BY 4.0), which means that the manuscript, images, and Supporting Information files will be freely available online, and any third party is permitted to access, download, copy, distribute, and use these materials in any way, even commercially, with proper attribution. For these reasons, we cannot publish previously copyrighted maps or satellite images created using proprietary data, such as Google software (Google Maps, Street View, and Earth). For more information, see our copyright guidelines: http://journals.plos.org/plosone/s/licenses-and-copyright.

a. You may seek permission from the original copyright holder of Figures 1,3,4,5,s4,s5,s6,s7,s8,s9,s10,s11,s12,s13,s14,s15,s16,s17,s18 and s19 to publish the content specifically under the CC BY 4.0 license. 

Additional Editor Comments:

Manuscript observe serious issues and require revision before considering it for publications.

Some major other concerns are:

1) Axis x and y are not properly labelled with units.

2) Study needs some statistical analysis to justify results and suggested to use some model.

3) Language needs to be proofread

Reviewers' comments:

Reviewer's Responses to Questions

**Comments to the Author**

1. Is the manuscript technically sound, and do the data support the conclusions?

Reviewer #1: Yes

Reviewer #2: Yes

2. Has the statistical analysis been performed appropriately and rigorously? 

Reviewer #1: Yes

Reviewer #2: Yes

3. Have the authors made all data underlying the findings in their manuscript fully available?

Reviewer #1: Yes

Reviewer #2: Yes

4. Is the manuscript presented in an intelligible fashion and written in standard English?

Reviewer #1: Yes

Reviewer #2: Yes

5. Review Comments to the Author

Reviewer #1: I found this topic to be extremely interesting, and I was very impressed with the way in which

the authors presented their findings. Following are a few minor observations and comments

that may help to further enhance your research work. The authors should address the

following points in detail.

_ The abstract should discuss the paper achievement.

_ Introduction of paper should include the contribution more prominently.

_ A “Literature Review” section/subsection should be added in the paper. The literature

review section should include a tabular comparison of existing models to identify their

shortcomings and strengths. The literature section is lacking in enclosure of formulation and

pollution aspect and requires more recent studies to be reviewed, therefore I suggest you cite

2022 and 2021 papers including the following works:

https://doi.org/10.1016/j.chemosphere.2021.132569

https://doi.org/10.1155/2022/6430120

http://dx.doi.org/10.36785/jaes.111501

https://doi.org/10.15244/pjoes/148065

https://doi.org/10.3389/fenvs.2022.945628

-Define the preliminary in the methodology section to explain all the mathematical notations

and symbols used in the paper.

-It is necessary to improve the quality of the figures.

-The authors should consider the comparison of results with existing studies.

- The authors should consider serious revision of formatting.

Reviewer #2: The present study we design is to find out the socioeconomic impacts of dust storms in China. I appreciate the general set-up of the experiment, but I also outline concerns below in the detailed comments that need attention before the study can be published.

Please don’t put title words in keywords.

Please provide policy recommendations on how to prevent these populations from the dust.

Introduction

Please increase the introduction to the following related literature.

https://www.sciencedirect.com/science/article/pii/S0045653521030411

https://www.mdpi.com/2073-4433/12/10/1338

Discussion

Better add similar heading as you put in the result.

The discussion part needs to improve by using previous studies.

Please start each paragraph with your finding and then a possible reason for this finding and compare it with previous research. I think plenty of literature is available in this regard. You can also use air pollution research in this regard.

The discussion part has only two references. Please include more references and enrich your discussion.

Please add the implication of your study.

6. PLOS authors have the option to publish the peer review history of their article (what does this mean?). If published, this will include your full peer review and any attached files.

Reviewer #1: No

Reviewer #2: **Yes: **Mir Muhammad Nizamani

---

## [Author Response · Author response to Decision Letter 0]

5 Jan 2023

Please see the attached response letters.

---

## [Decision Letter · Decision Letter 1]

12 Apr 2023

PONE-D-22-31637R1The Population Affected by Dust in China in the SpringtimePLOS ONE

Dear Dr. Zhang,

Thank you for submitting your manuscript to PLOS ONE. After careful consideration, we feel that it has merit but does not fully meet PLOS ONE’s publication criteria as it currently stands. Therefore, we invite you to submit a revised version of the manuscript that addresses the points raised during the review process.

We look forward to receiving your revised manuscript.

Kind regards,

Yangyang Xu

Academic Editor

PLOS ONE

Reviewers' comments:

Reviewer's Responses to Questions

**Comments to the Author**

1. If the authors have adequately addressed your comments raised in a previous round of review and you feel that this manuscript is now acceptable for publication, you may indicate that here to bypass the “Comments to the Author” section, enter your conflict of interest statement in the “Confidential to Editor” section, and submit your "Accept" recommendation.

Reviewer #1: All comments have been addressed

Reviewer #2: All comments have been addressed

Reviewer #3: (No Response)

Reviewer #4: (No Response)

2. Is the manuscript technically sound, and do the data support the conclusions?

Reviewer #1: Yes

Reviewer #2: Yes

Reviewer #3: Yes

Reviewer #4: Partly

3. Has the statistical analysis been performed appropriately and rigorously? 

Reviewer #1: Yes

Reviewer #2: Yes

Reviewer #3: Yes

Reviewer #4: No

4. Have the authors made all data underlying the findings in their manuscript fully available?

Reviewer #1: Yes

Reviewer #2: Yes

Reviewer #3: Yes

Reviewer #4: (No Response)

5. Is the manuscript presented in an intelligible fashion and written in standard English?

Reviewer #1: Yes

Reviewer #2: Yes

Reviewer #3: Yes

Reviewer #4: Yes

6. Review Comments to the Author

Reviewer #1: While the authors have addressed my concerns, there is always room for improvement. Moreover, I am curious as to why the authors did not make these changes at the time of submission. It was very impressive how the authors presented their findings. It is greatly appreciated that the authors addressed my concern. There are no further questions on my part.

Reviewer #2: (No Response)

Reviewer #3: This paper, a revision of a manuscript I didn’t initially review, presents a really nicely-delivered intersection of two large datasets (the MODIS dust product, and population and socioeconomic data) over nearly two decades, and yields useful conclusions on the degree of human exposure to dust hazards over China.

Previous reviewers’ comments seem to have been well addressed in this revision, and I recommend its publication after a few relatively minor comments are addressed:

L35 Delete ‘Still’ – it does not follow logically

L43 “A previous study…”

L62-66 I appreciate that this section was added in direct response to a comment raised by both reviewers, but for me, it sits awkwardly here in the way it is worded. It reads, effectively as a set of conclusions at the end of the introduction, and comes immediately after the rationale for the study, which works very well in lines 58-62. Can these lines be altered to set these more as goals or ideas to be explored rather than as a fait accompli.

L70 Suggest “MODIS (Moderate Resolution Imaging Spectroradiometer) instruments are aboard the Terra and Aqua satellites and..”

L114 and Figure 1 We are told here that the mean DOD in Xinjiang is 0.36 over the full time series, but it seems to me equally relevant to note that in southern half of Xinjiang in the Taklamakan it is more like >0.6. Averaging at a province level here seems to me to miss the point rather.

L144-145 Worth noting here than 2004 is the prime example here?

L181 Refer to Fig. 1a at mention of the Heihe-Tengching line

L251-252 Specific mention of ‘visual inspection’ being inconclusive suggests to the reader than something else was/could be done to further explore this. Reword, or be more definitive that there is no clear relationship. Also, mention here of dust “generating” inequalities suggests a definite causal relationship. Might it not be that dust is generated in/affects rural areas more, which are poorer; this would be a correlation, not a causation.

L313 ‘pullulation’ = “population”, I assume?

L350 “on the order of millions”

L353 “it is crucial to understand the future trend” – this has not been done in this study, and in fact you’ve just spend about eight lines explaining why this would be very difficult/impossible to do accurately. Why then just state that this is crucial?

Reviewer #4: This study presents the spatiotemporal variations of population affected by dust in China and investigates the inequalities in dust impacts on different socioeconomic groups. To my best knowledge, there are not many studies on this topic, although the dust-affected regions and population are reported for some dust events. Most of the manuscript is clearly presented and well written. I have some comments for further consideration.

Major comments;

Section 4.1, in particular “more useful” in Line 289: Although I agree there are some advantages in DOD compared to surface records, there are at least two caveats in DOD for assessing the impacts of dust-affected population. First, although monthly DOD is provided, this does not represent the real distribution of dust, as some dust distributions cannot be distinguished by satellites due to the cloud cover and they are often represented as missing data. Second, DOD represents the dust in the total column, while humans are only exposed to the dust particles near the surface. There is some relationship between surface dust concentration and DOD, but there is some uncertainty. Although this work is useful and a good reference for the community, the authors should not overstate their work and should be aware of its caveats.

Lines 120-121: Description of dust emission and transportation is not accurate: Please be aware that dust particles in Taklimakan Deserts are mostly trapped within the Tarim Basin, while the dust originating from Gobi Deserts is mainly transported to downwind regions including North China and East China.

Lines 123-124: Not only dust from eastern Inner Mongolia but also dust from Gobi Deserts.

Lines 336-338: This sentence is misleading. Similar forecast models for predicting East Asia dust are already available to provide the forecast for the public.

Specific comments:

Line 43: The paragraphs are not well edited and it is a bit difficult to find the beginning of the paragraph. One blank row is needed between this paragraph and the previous paragraph.

Line 88, Section 2.3: To facilitate the explanation, I suggest the authors provides the figure in accompany with the equation for description (probably in supplemental file).

Line 118: central China -> North China?

Line 127: It is difficult to find the orders of figure in the appendix.

Lines 131-132: I cannot find the reference. I think it may be the Annual Book of Meteorological Disasters in China for 2018 (2019), as the records ended in 2018.

Lines 199-200: the Municipality of Tianjin -> Tianjin?

Line 284: meteorological factors: It is not clear to me.

Line 314: 142 million: I think the number can not be directly compared to that in Liu et al. (2007), as different criteria are used between this study and that study.

Lines 323-325: Inequalities indicate the difference between poor and wealthy peoples, but they do not imply that healthcare does not improve in past decades. Please clarify there is not missing leading information.

Lines 367-369: people … affected: It is not clear to me.

Line 370: "risk management" is not clear to me.

7. PLOS authors have the option to publish the peer review history of their article (what does this mean?). If published, this will include your full peer review and any attached files.

Reviewer #1: No

Reviewer #2: **Yes: **Mir Muhammad Nizamani

Reviewer #3: No

Reviewer #4: No

---

## [Author Response · Author response to Decision Letter 1]

1 Jun 2023

See the attached response letter.

---

## [Decision Letter · Decision Letter 2]

6 Jul 2023

The Population Affected by Dust in China in the Springtime

PONE-D-22-31637R2

Dear Dr. Zhang,

We’re pleased to inform you that your manuscript has been judged scientifically suitable for publication and will be formally accepted for publication once it meets all outstanding technical requirements.

Kind regards,

Yangyang Xu

Academic Editor

PLOS ONE

Additional Editor Comments (optional):

Reviewers' comments:

Reviewer's Responses to Questions

**Comments to the Author**

1. If the authors have adequately addressed your comments raised in a previous round of review and you feel that this manuscript is now acceptable for publication, you may indicate that here to bypass the “Comments to the Author” section, enter your conflict of interest statement in the “Confidential to Editor” section, and submit your "Accept" recommendation.

Reviewer #3: All comments have been addressed

Reviewer #4: All comments have been addressed

2. Is the manuscript technically sound, and do the data support the conclusions?

Reviewer #3: Yes

Reviewer #4: Yes

3. Has the statistical analysis been performed appropriately and rigorously? 

Reviewer #3: Yes

Reviewer #4: Yes

4. Have the authors made all data underlying the findings in their manuscript fully available?

Reviewer #3: Yes

Reviewer #4: Yes

5. Is the manuscript presented in an intelligible fashion and written in standard English?

Reviewer #3: Yes

Reviewer #4: Yes

6. Review Comments to the Author

Reviewer #3: (No Response)

Reviewer #4: (No Response)

7. PLOS authors have the option to publish the peer review history of their article (what does this mean?). If published, this will include your full peer review and any attached files.

Reviewer #3: No

Reviewer #4: No

---

## [Editor Report · Acceptance letter]

24 Jan 2023

PONE-D-22-31637R1 

The Population Affected by Dust in China in the Springtime 

Dear Dr. Zhang:

I'm pleased to inform you that your manuscript has been deemed suitable for publication in PLOS ONE. Congratulations! Your manuscript is now with our production department. 

Kind regards, 

on behalf of

Dr. Uzair Aslam Bhatti 

Academic Editor

PLOS ONE